# Automatic Building Extraction from Google Earth Images under Complex Backgrounds Based on Deep Instance Segmentation Network

**DOI:** 10.3390/s19020333

**Published:** 2019-01-15

**Authors:** Qi Wen, Kaiyu Jiang, Wei Wang, Qingjie Liu, Qing Guo, Lingling Li, Ping Wang

**Affiliations:** 1National Disaster Reduction Center of China, Beijing 100124, China; whistlewen@aliyun.com (Q.W.); wangwei@ndrcc.gov.cn (W.W.); lilingling@ndrcc.gov.cn (L.L.); wangping@ndrcc.gov.cn (P.W.); 2School of Computer Science and Engineering, Beihang University, Beijing 100191, China; kyjiang@buaa.edu.cn; 3The People’s Insurance Company of China, Beijing 100022, China; guoqing@picc.com.cn; 4The State Key Laboratory of Virtual Reality of Technology and Systems, Beihang University, Beijing 100191, China

**Keywords:** building extraction, deep learning, Mask R-CNN, rotation bounding box, receptive field block, instance segmentation

## Abstract

Building damage accounts for a high percentage of post-natural disaster assessment. Extracting buildings from optical remote sensing images is of great significance for natural disaster reduction and assessment. Traditional methods mainly are semi-automatic methods which require human-computer interaction or rely on purely human interpretation. In this paper, inspired by the recently developed deep learning techniques, we propose an improved Mask Region Convolutional Neural Network (Mask R-CNN) method that can detect the rotated bounding boxes of buildings and segment them from very complex backgrounds, simultaneously. The proposed method has two major improvements, making it very suitable to perform building extraction task. Firstly, instead of predicting horizontal rectangle bounding boxes of objects like many other detectors do, we intend to obtain the minimum enclosing rectangles of buildings by adding a new term: the principal directions of the rectangles *θ*. Secondly, a new layer by integrating advantages of both atrous convolution and inception block is designed and inserted into the segmentation branch of the Mask R-CNN to make the branch to learn more representative features. We test the proposed method on a newly collected large Google Earth remote sensing dataset with diverse buildings and very complex backgrounds. Experiments demonstrate that it can obtain promising results.

## 1. Introduction

During the last 10 years, many countries in the world have suffered from natural disasters, which are increasing in frequency and intensity and brought huge loss in exposure of persons and assets [1]. Disaster loss assessment can provide technical support and decision-making basis for disaster relief and post-disaster reconstruction. Loss from building damage always accounts for a high percentage of all losses especially in typhoon, earthquake, flood, and geological disasters, so loss assessment of building damage is obviously an essential work of the whole loss assessment. Remote sensing images have played an important role in building damage assessment for their characteristics of wide coverage, high resolution and high time efficiency [2,3]. Building footprint vector data can provide basic information of buildings, such as outline, area, shape. It can be used to assess the damage condition of each building and calculate the total number of damaged buildings in a disaster region [4].

How to extract building footprint from pre-disaster high resolution remote sensing images is a key problem which has brought much attention from both academia and industry for decades. In consideration of feature extraction, the classical methods of building footprint extraction from remote sensing images can be summed up into two categories: bottom-up data-driven methods and top-down model-driven methods [5]. The bottom-up data-driven methods mainly consider the low-level features such as lines, corners, regional textures, shadows, and height differences from remote sensing images, and assemble them under some rules to identity building targets. The top-down model-driven methods start from the semantic model and prior knowledge of the whole building targets, use some high level global features, and guide the optimization of element extraction, image segmentation, spatial relationship modeling, contour curve evolution to be close to the building targets. The accuracy and efficiency of these two methods are usually difficult to meet the practical application requirements of remote sensing building extraction.

Recent years, the deep learning methods, represented by Convolutional Neural Network (CNN) and Recurrent Neural Network (RNN), have been gradually dominating the fields of artificial intelligence [6]. The progress in deep learning has greatly promoted the developments of image classification (AlexNet [7], Visual Geometry Group Network (VGG) [8], GoogleNet [9], Residual Network (ResNet) [10]), image semantic segmentation (Fully Convolutional Network (FCN) [11], U-Net [12]), object detection (Region Convolutional Neural Network (R-CNN) [13], Fast R-CNN [14], Faster R-CNN [15], Region Fully Convolutional Network (R-FCN) [16], You Only Look Once (YOLO) [17], Single Shot Multibox Detector (SSD) [18]), and some other classical computer vision problems. These great achievements have inspired researchers in remote sensing community to apply deep learning techniques to building extraction. A straightforward strategy is adapting these algorithms to building extraction task. For instance, Vakalopoulou et al. [19] presented one of the first deep learning based building extraction methods, in which a small patch was fed into AlexNet and Support Vector Machine (SVM) assembling classifier to detect buildings, then the pixel-wise classification result was refined by Markov Random Filed (MRF). They achieved 90% average correctness in Quickbird, Worldview images. In [20], a 3-layer hierarchically fused fully convolutional network (HF-FCN) was developed to test on Massachusetts Buildings Dataset [21] and achieved 91% average recall rate, which was based on FCN, truncated the 6th, 7th fully connected layers and 5th pooling layer, fused multi-level convolution, upper sampling layers and formed a cascaded segment network. Zhang et al. [22] developed a CNN based building detection method. They employed a multi-scale saliency map to locate building-up areas, and combined with sliding window to obtain the candidate patches classified by CNN. A promising result of 89% detection precision was achieved on Google Earth remote sensing images with a spatial resolution of 0.26 m. A network structure, which adopted the SegNet [23] like encoder-decoder pair style and incorporated the up-sampling and densification operations into deconvolution layers, is proposed and achieved 95.62% building segmentation precision in ISPRS 2D Semantic Labelling Challenge [24]. A patch-based CNN classification network is proposed and has an accuracy of building segmentation exceeding that of previous similar work in Massachusetts Buildings Dataset and Abu Dhabi Dataset, which uses a Global Average Pooling layer instead of the full connection layer and uses the super pixel division (SLIC) for post-processing [25]. A multi-constraint fully convolutional networks (MC–FCNs) is proposed and achieves a 97% total accuracy of building segmentation in New Zealand aerial remote sensing images, which adopts the basic structure of a fully skip connected U-Net and adds a multi-layer constraint between each feature map layer and the corresponding multi-scale true value annotation data, increasing the expression ability of the middle layer [26]. Based on the architecture of U-Net, the Res-U-Net segmentation network is established, which uses residual units instead of plain neural units as basic blocks, and the guided filter is then adopted as subsequent step to fine-tune the building extraction result. The segmentation accuracy of the building on the ISPRS 2D Semantic Labelling Challenge dataset is superior to 97% [27]. In general, the existing work of deep learning method for building extraction from high-resolution remote sensing images is mainly based on semantic segmentation, and the work on target detection and image classification method is few. The main idea is to improve the context information by adding the multi-layer features to the FCN framework, and to improve the ability to adapt to the complex background of remote sensing images and the small building targets.

Semantic segmentation under complex geospatial backgrounds is likely to result in edge connection among closely adjacent buildings, which is unfavorable for subsequent edge extraction and outline fitting owing to edge confusion of buildings. Mask R-CNN [28], a pioneer work in instance segmentation which is a task predicting bounding boxes and segmentation masks simultaneously, have achieved significant improvement. In this work, segmentation is carried out based on detection result, making it especially suitable to deal with outline extraction of densely distributed small buildings. A newly defined Rotation Bounding Box (RBB), which involves angle regression, is incorporated into Faster R-CNN framework, and this method forces the detection networks to learn the correct orientation angle of ship targets according to angle-related IoU and angle-related loss function [29]. Meanwhile, a novel Receptive Field Block (RFB) module, which makes use of multi-branch pooling with varying kernels and atrous convolution layers to simulate RFs of different sizes in human visual system, is developed to strengthen the deep features learned from lightweight CNN detection models [30]. In this paper, we incorporate the RBB and RFB into the RPN stage and segmentation branches of Mask R-CNN framework, respectively. This improvement can provide a denser bounding box, and furtherly promote the accuracy of mask prediction owing to better adaptation to multi-scale building targets. 

The main contributions of this paper include:Different from previous FCN-based methods, instance segmentation framework is applied into building detection and segmentation, which can better deal with closely adjacent small buildings and some other tough problems.We adapt rotatable anchors into the RPN stage of Mask R-CNN framework, which regress a minimum area bounding rectangle (MABR) likely rotation bounding box and eliminate redundant background pixels around buildings.We use several RFB modules to boost the segmentation branch of Mask R-CNN framework, which can better accommodate to multi-scale building targets by parallel connecting multi-branch receptive fields with varying eccentricities.

Experiments based on a newly collected large building outline dataset show that our method, improved from Mask R-CNN framework, has a state-of-the-art performance in joint building detection and rooftop segmentation task. 

The remainder of this paper is organized as follows: Section 2 presents the details of building extraction method; Section 3 describes the experimental results in Google Earth remote sensing dataset; Section 4 is a discussion of our method and some possible plan of improvements; Section 5 presents our concluding remarks.

## 2. Methods of Building Extraction from Remote Sensing Images

Similar to Mask R-CNN, our method mainly consists of four parts. Firstly, rotation anchors are introduced into RPN stage since we intend to predict the minimum area bounding rectangle of buildings. Secondly, the feature maps of ROIs are rotated anticlockwise into horizontal rectangles and are then processed by ROI Align. Thirdly, the regression branch revises the coordinate of bounding box, the classification branch predicts the corresponding classification scores, and the segmentation branch produces corresponding object masks through several RFB modules. Finally, the bounding box and mask are rotated clockwise by the regressed angle as the instance segmentation results. The losses of the three branches are computed and summed to form a multitask loss. Figure 1 illustrates the schematic architecture of the proposed method.

### 2.1. Rotation Region Proposal Network

Feature map from backbone is fed into rotation region proposal network. In the learning stage, the rotation bounding box is defined as the ground truth of each building sample for detection. Rotation proposals are formulated by adding angle parameter, and are generated by traversing every composition of ratio, scale and angle. In the prediction stage, the feature maps of rotation detection bounding boxes generated by rotation RPN are rotated anticlockwise to horizontal rectangles by the regressed angle. Then after ROI Align, they are transferred to the multi-branch network.

#### 2.1.1. Rotation Bounding Box

The refined outline of each building is regarded as the ground truth data, which is necessary for segmentation task. However, for the detection purpose, the ground truth is the minimum area bounding rectangle (MABR) of buildings. Unlike traditional horizontal bounding rectangle, MABR is a dense bounding box, which has the minimum area among all of the bounding rectangles and normally is inclined from horizontal axis. Figure 2 illustrates the outline and the minimum area bounding rectangle of buildings. So 5 parameters, i.e., (x,y,w,h,θ) are used to represent the rotation bounding box, where (x,y) represent the center coordinate of bounding box, (w,h) represent the length of the short side and the long side of the bounding box respectively, and the angle between the long side of MABR and *x*-axis rotated from counterclockwise direction is represented as parameter θ. θ is constrained within the interval [−π/4, 3π/4) to ensure the uniqueness of MABR. Figure 3 presents angle parameter θ of rotation bounding box.

#### 2.1.2. Rotation Anchor

In order to match the rotation bounding box, rotation anchors are designed by adding rotation angle to traditional anchor parameters. Buildings account for different functions, such as factory and residence. Factory buildings, housings for urban and rural residents, office buildings are likely to have distinct aspect ratios. According to statistics of a large amount of building samples, we set the aspect ratios as {1:2, 1:3, 1:5, 1:7}. Six scales, i.e., {8, 16, 32, 64, 128, 256}, are kept to fit in the scale variation of buildings. In addition, we adopt six orientations {−π/6, 0, π/6, π/3, π/2, 2π/3} to adjust anchors to match angle changes of buildings. 144 rotation anchors (4 aspect ratios, 6 scales, 6 orientations) will be created for each pixel on the feature map, 720 outputs (5 × 144) for the reg layer and 384 score outputs (2 × 144) for the cls layer.

#### 2.1.3. Leveling ROIs

The rotation ROIs output from the RPN stage always have a certain angle against horizontal axis represented by parameter θ. The feature map of ROI is rotated by the θ angle anticlockwise around its center into a horizontal rectangle of the same size by bilinear interpolation. The transformed coordinates can be calculated as follows:(1)$$\left[\begin{array}{c} x^{\prime \prime }\\ y^{\prime \prime } \end{array}\right]=\left[\begin{array}{cc} \cos\theta & \sin\theta \\ -\sin\theta & \cos\theta \end{array}\right]\left[\begin{array}{c} x^{\prime }-x\\ y^{\prime }-y \end{array}\right]+\left[\begin{array}{c} x\\ y \end{array}\right]$$
where (x,y) represent the center coordinate of bounding box, $\left(x^{\prime },y^{\prime }\right)$ represent the coordinate of pixel in original ROI feature map, $\left(x^{\prime \prime },y^{\prime \prime }\right)$ represent the coordinate of pixel in transformed ROI feature map. Then we use the ROI Align to process the horizontal feature maps of ROIs and transfer the resulting fixed-size feature maps to the following multi-branch prediction network.

### 2.2. Multi-Branch Prediction Network

Multi-branch prediction network has three branches: two branches perform classification and bounding-box regression respectively, the third branch performs segmentation and generates masks. The segmentation branch is reconfigured with Receptive Field Block modules to obtain finer masks by integrating advantages of inception block and atrous convolution. Then, the regressed bounding-box and the predicted mask are simultaneously rotated to their original θ angle obtained from RPN stage. In this way, we can obtain the final instance segmentation results of buildings.

#### 2.2.1. Receptive Field Block

The scales of buildings vary significantly, ranging from a dozen of pixels to thousands of pixels. To better handle scale viability problem, a new architecture named Receptive Field Block is built upon the structure of Inception-ResNet module [31] by replacing the filter concatenation stage of Inception V4 module with residual connection and stacking atrous convolution of different kernel sizes and sampling rates. Figure 4 shows the architecture of a RFB module. The 1 × 1 atrous convolution with rate 1, 3 × 3 atrous convolution with rate 3 and 3 × 3 atrous convolution with rate 5 are inserted into a paralleled three-branch structure respectively, and the feature maps extracted for different sampling rates are further concatenated and followed by a 1 × 1 convolution. The output of the filter is then residual added with the output of pre-stage layer by a shortcut channel. Each branch of the three-branch structure consists of a 1 × 1 convolution layer to decrease the number of channels in the feature map plus an n × n convolution layer.

#### 2.2.2. RFB Stacked Segmentation Network Branch

We replaced each convolution layer of the original segmentation branch of Mask R-CNN with the RFB module, and then activated with sigmoid function, as shown in Figure 5. Two RFB modules connected in sequence can enlarge the receptive field and avoid time-consuming. Accumulating more RBF blocks would slightly improve the performance. However, when attaching more than three blocks, it will lead to unstable accuracy and make training more difficult. The output map of this branch is the mask of building target.

#### 2.2.3. Inverse Rotation of Mask

The bounding box regression branch only revises the coordinates of horizontal rectangles, i.e., (x,y,w,h). The angle θ generated from the Rotation RPN stage is adopted as the final angle parameter. The horizontal rectangle is rotated clockwise by the θ angle as the final rotation bounding box. The m × m mask output predicted from the segmentation branch is first rotated clockwise by the θ angle, and then is resized to the size of final bounding box and binarized at a threshold of 0.5.

### 2.3. Loss Function

The positive labels are assigned to the anchors as follows: (i) the anchor/anchors with the highest IoU overlap with a ground-truth box; (ii) an anchor which has an IoU overlap higher than 0.8 and an angular separation less than 10 degrees with the ground-truth box. The Negative label is assigned to the anchor following two conditions: (i) an anchor has an IoU overlap less than 0.2; (ii) an anchor has an IoU overlap higher than 0.8 but has an angular separation higher than 10 degrees. Other anchors without positive or negative labels are not considered during training.

We follow the multi-task loss of Mask R-CNN which is defined as follow to train our method:(2)$$L(p_{i}^{*},p_{i},t_{i}^{*},t_{i},s_{i}^{*},s_{i})=\frac{1}{N_{cls}}\sum_{i}L_{cls}(p_{i}^{*},p_{i})+\lambda \frac{1}{N_{box}}\sum_{i}p_{i}^{*}L_{box}(t_{i}^{*},t_{i})+\gamma \frac{1}{N_{mask}}\sum_{i}L_{mask}(s_{i}^{*},s_{i})$$
where $p_{i}^{*}$ represents the ground-truth label of the object, $p_{i}$ is the predicted probability distribution of anchor i being an object of different classes, $t_{i}^{*}$ is the vector representing the coordinate offset of ground-truth box and positive anchors, $t_{i}$ represents the offset of the predicted five parameterized coordinate vector and that of ground-truth box, $s_{i}^{*}$ is the matrix of ground-truth binary mask, $s_{i}$ represents the predicted mask of the object. The hyper-parameter λ and γ in Equation (2) controls the balance between the three task losses.
(3)$$L_{cls}(p^{*},p)=-\log p^{*}p$$
(4)$$L_{box}(t_{i}^{*},t_{i})=smooth_{L_{1}}(t_{i}^{*}-t_{i})$$
(5)$$smooth_{L_{1}}(x)=\left\{\begin{array}{c} 0.5x^{2},if\left|x\right|<1\\ \left|x\right|-0.5,otherwise \end{array}\right\}$$
(6)$$L_{mask}(s^{*},s)=-(s^{*}\log(s)+(1-s^{*})\log(1-s))$$

The regression mode for 5 coordinate parameters of rotational bounding box is defined as follow:(7)$$t_{x}=(x-x_{a})/w_{a},t_{y}=(y-y_{a})/h_{a},\;t_{w}=\log(w/w_{a}),t_{h}=\log(h/h_{a}),\;t_{\theta }=\theta -\theta_{a}+k\pi$$
(8)$$t_{x}^{*}=(x^{*}-x_{a})/w_{a},t_{y}^{*}=(y^{*}-y_{a})/h_{a},\;t_{w}^{*}=\log(w^{*}/w_{a}),t_{h}^{*}=\log(h^{*}/h_{a}),\;t_{\theta }^{*}=\theta^{*}-\theta_{a}+k\pi$$
where x, y, w and h denote the box’s center coordinates and its width and height. Variables x, $x_{a}$ and $x^{*}$ are for the predicted box, anchor box, and ground-truth box respectively; the same is for y, w, h and θ. The parameter k∈Z to keep $t_{\theta }$ and $t_{\theta }^{*}$ in the range [−π/4, 3π/4).

## 3. Results

### 3.1. Data and Research Area

To assess the performance of the proposed method and facilitate future researches, we collected a large volume of images from Google Earth in Fujian province, China, as shown in Figure 6. Diverse regions including urbans, towns, and villages are selected to cover different kinds of buildings. Several examples are shown in Figure 7, from which we can see almost all types of buildings from village to urban, from small rural housing to villa and high-rise apartment, from L shape to U shape are included in our dataset, providing us plenty of samples for training the models. 86 typical regions of spatial resolution 0.26 m are selected, with the image size ranging from 1000 × 1000 to 10,000 × 10,000 pixels. After obtain the images, 5 students major in Geography and surveying science were asked to label the buildings with polygon vector using ArcGIS 10.2 manually. The polygon vectors fit to the outlines of building footprints, as shown in Figure 7. Because deep learning models can only learn parameters from fixed-size images with numerical labels, we crop the images into 500 × 500, and map the vector boundaries into bounding boxes. Finally, we have 2000 images and about 84,366 buildings in total. We split the dataset equally into two parts, one for training and the other one for testing.

### 3.2. Implementation Details

The model is built upon Mask R-CNN framework. We use PyTorch to implement the proposed method and train it with Adam optimizer. The backbone of the model is ResNet-101 which was pre-trained on ImageNet dataset. The learning rate was initialized with 0.001 and decayed in every 25 k iterations. It will converge in 80 k iterations. Other hyperparameters such as weight decay and momentum were set as 0.0001 and 0.9 as recommended. At inference time, 500 proposals are generated for predicting buildings and refine their locations. The top 100 predictions with the highest scores are sent to the segmentation task branch and obtain their masks. All experiments including training and testing of models are conducted on a single 1080Ti GPU with 12 GigaByte memory on board.

### 3.3. Evaluation of Detection Task

Building detection from very complex backgrounds is an important task. Detecting objects from images has been a hot research topic in computer vision community. And lots of deep learning based methods have been proposed in recent years. Most of these methods can be categorized into two groups: two-stage methods and one stage methods. Two-stage methods has an RPN network that generates candidate regions potentially containing objects and a followed network classifies these regions into different object categories and predicts their fine coordinates, simultaneously. The representative method is the Faster R-CNN and its variants. While one stage methods directly predict the classification score and coordinates of the objects from the feature maps without an RPN stage. Thus, one stage methods are faster than two-stage methods in inference however have poor performance in detecting and locating objects. In this work, we compare our method with Mask R-CNN and Faster R-CNN since they obtain the state-of-the-art results. Two different networks VGG [9] and ResNet101 [10] are utilized as a backbones of the Faster R-CNN. The proposed and Mask R-CNN are not configured with VGG network because Mask R-CNN and the proposed method are actually built upon Faster R-CNN, thus it is unnecessary to repeat the VGG configuration again. We use mean average precession (mAP) to evaluate the performance of the proposed method. The results are listed in Table 1. A few examples are shown in Figure 8.

From Table 1 we can see that Faster R-CNN configured with ResNet101 outperforms its VGG version significantly, indicating powerful ability of residual networks. ResNets have been utilized widely in various computer vision tasks and demonstrate superior performance than other shallower networks, such as VGG nets. Thus, in the following experiments we also believe in ResNets and employ ResNet101 which is probability the most widely used residual network as backbone of the proposed method. Mask R-CNN-ResNet101 obtains similar results to Faster R-CNN-ResNet101. They are actually the same model if only the detection task is considered. The proposed method improves the results with help of the rotation anchors. The reason behind this maybe that the rotated anchors provide more information of target characteristics (i.e., rotation angle) than normal anchors, so they are more suitable for capturing features of rotated objects. They have a higher possibility of filtering out pixels of distractive backgrounds than normal anchors which leads to better results.

From Figure 8 it can be observed that Faster R-CNN configured with VGG miss detect buildings at most. The image in the first row is a very challenging one. There are many buildings locating closely and are very hard to distinguish from each other. Faster R-CNN-VGG miss many buildings, while Mask R-CNN-ResNet101 and the proposed method obtain the best results though there are still many missing buildings. The rotated bounding boxes fit bounding footprints well, as can be seen from the last column of Figure 8.

### 3.4. Evaluation of Segmentation Task

Segmenting buildings from their surrounding backgrounds also known as building extraction. In this subsection, we compare our method with segmentation branch of Mask R-CNN. Three indicators including precision, recall and *F*_1_ score are used to evaluate the performance of the proposed method. We report them in Table 2 and shown some examples of the segmentation results in Figure 9.

From Table 2 we can see that the proposed method outperforms the Mask R-CNN-ResNet101 in terms of all of the three indicators. One should know that the Fujian dataset is very challenging. Many of the buildings are hard to distinguish from surroundings due to the poor quality of the Google Earth images. The RFB block [30] inspired by the mechanism of the human visual systems plays a central role in improving the performance of the segmentation. One possible explanation may be that the atrous convolution enlarges the receptive fields and combinations of different radius and rates enable extractions of more powerful features. This can be read from Figure 9, from which we can see the proposed method successfully segment some indistinguishable buildings from backgrounds which are missed by Mask R-CNN-ResNet101, as can be seen from the second and forth columns of Figure 9.

Mask R-CNN produce instance level segmentation results, which means different instances of the same category are annotated with distinct pixel-level labels, as indicated by the different colors of Figure 9. Instance segmentation is extremely useful when buildings close to each other with adjacent boundaries or even share with the same wall. General segmentation methods such as U-Net-style networks [32] cannot distinguish different instances. Thus, for adjacent buildings they could generate one big mask for several buildings. Mask R-CNN provides a good solution to this by segmenting buildings in their bounding boxes. This can also help to improve accuracy of segmentation and provide a fine outline of buildings. We demonstrate that the results could be further boosted by inserting the RFB blocks.

## 4. Discussion

Our proposed method has achieved improved performance for building extraction and segmentation tasks in terms of quantitative indicators, especially on building detection. However, we believe the performance could further be improved from the following aspects. More and diversity building samples. Deep neural networks are data hungry models, requiring a huge volume of training samples. Although we have labeled thousands of buildings to train our network, providing more samples will further boost the performance. In addition, buildings have diversity sizes, structures. For instance, factory buildings and residential houses possess distinctly different features. Even residential houses, buildings from city and village are with different sizes, aspect ratio and shapes. To detect them all, samples should cover as many instances as possible. Moreover, complex backgrounds could be distractions to the detector, especially there are objects with similar appearance, such as vehicle, ships, roads, and so on. An example is shown in Figure 10. It is better to label a certain amount of buildings under complex backgrounds.Refine rotated angle of bounding box. In this work, the value of a rotated angle is regressed from the RPN stage. Since there is only one category i.e., buildings to be detected, ROIs generated by the RPN network should be close to that of detection branch, thus we use angles from the RPN as the final rotation angles. However, we believed that, similar to bounding boxes regression they can be further refined by the second stage. In future, we will focus on two solutions. The first one is designing a new branch and adding it after the second stage to refine the rotated angle. The new branch will accept the rotated mask as input and predict the angle. The second one is transforming the vertical ROIs generated by RPN to the second stage. The vertical ROIs consist of rotation information thus can be used to infer the angle value. Since ROI Align is applied in the RPN stage, we will obtain more accurate angles.Network compression. Mask R-CNN framework has a huge number of parameters, which will consume large amount of computation resource and lead to decrease in inference time. In recent years, with the rapid development of mobile device and the demand for real-time computation, researchers have attempted to compress the size of the deep models while maintaining their performance. These methods resolve the network compression problem from three aspects: designing light network, network pruning and kernel sparsity. Both the backbone of the Mask R-CNN and the proposed method are based on residual network, which could be pruned to produce a lighter backbone. In addition to this, some inborn light networks such as ShuffleNet [33], CornerNet [34] can be used to design the proposed method. 

Building extraction is still an open problem requiring more research efforts. In the future, we will plan to design and train specific network aiming at detecting closely located small buildings, large scale buildings, buildings with special shapes and under confusing backgrounds.

## 5. Conclusions

In this paper, we propose an automatic building extraction method based on improved Mask R-CNN framework, which detect the rotated bounding boxes of buildings and segment them from very complex backgrounds, simultaneously. The rotation anchor with inclined angle is used to regress the rotation bounding box of buildings in the RPN stage. Then, after rotation anticlockwise and ROI Align, feature maps are transferred to the multi-branch prediction network. In addition, RFB modules are inserted to the segmentation branch to handle multi-scale variability, and other branches output the classification scores and horizontal rectangle coordinate. Finally, the mask and rectangle bounding box are rotated clockwise by the inclined angle as the final instance segmentation result. Experiment results on a newly collected large Google Earth remote sensing dataset with diverse buildings under complex backgrounds show that our method can achieve promising results. The future work can be focused on samples annotation, improvement, and compression of network structure to promote the performance of our method.

## Figures and Tables

**Figure 1 sensors-19-00333-f001:**
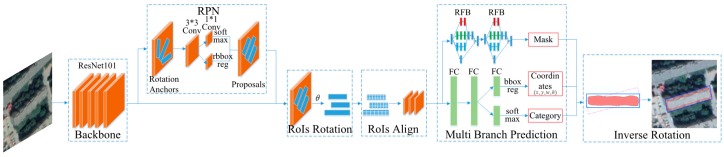
Schematic architecture of the proposed method.

**Figure 2 sensors-19-00333-f002:**
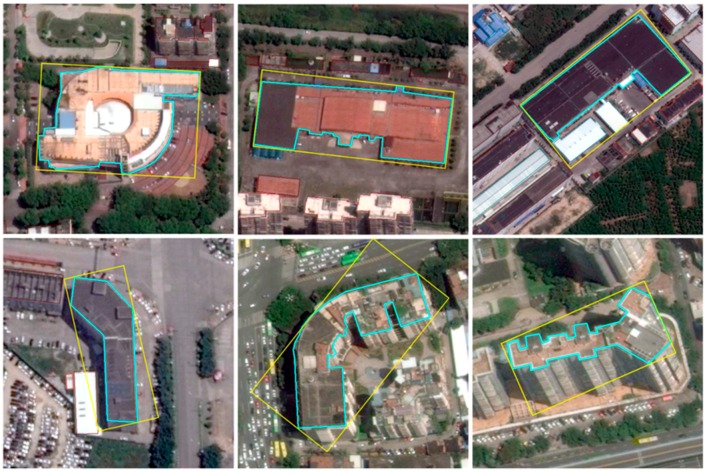
Outline and the minimum area bounding rectangle (MABR) of buildings. Blue lines are the outlines of the buildings and yellow lines are their MABR.

**Figure 3 sensors-19-00333-f003:**
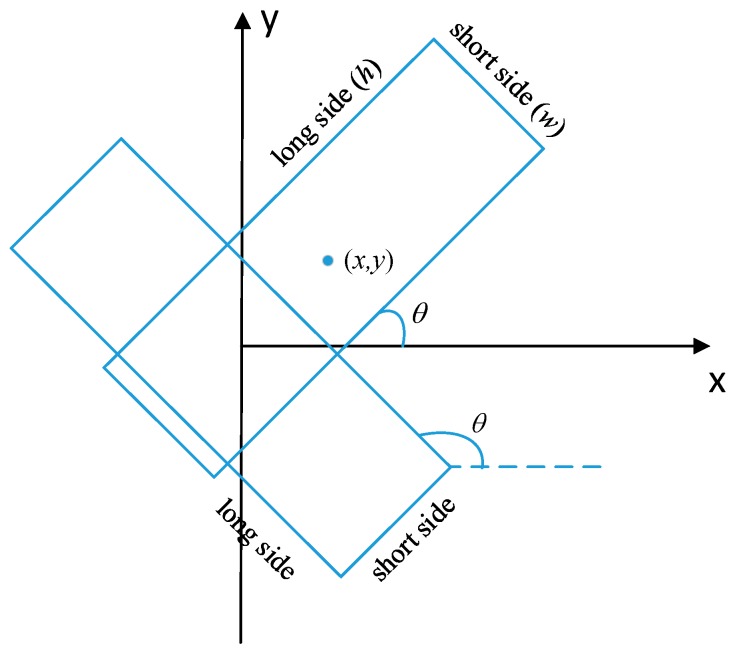
Angle parameter θ of rotation bounding box.

**Figure 4 sensors-19-00333-f004:**
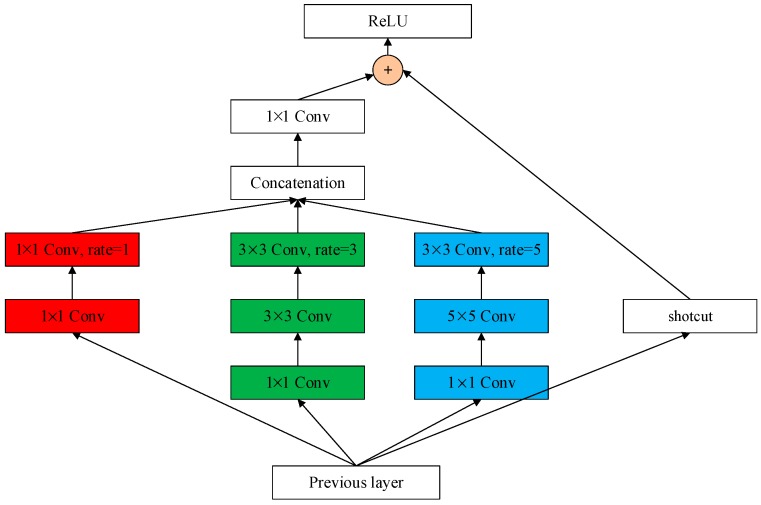
Structure of Receptive Field Block.

**Figure 5 sensors-19-00333-f005:**
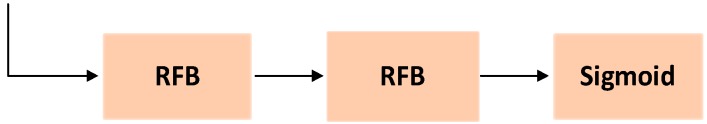
Pipeline of Receptive Field Block (RFB) modules stacked network.

**Figure 6 sensors-19-00333-f006:**
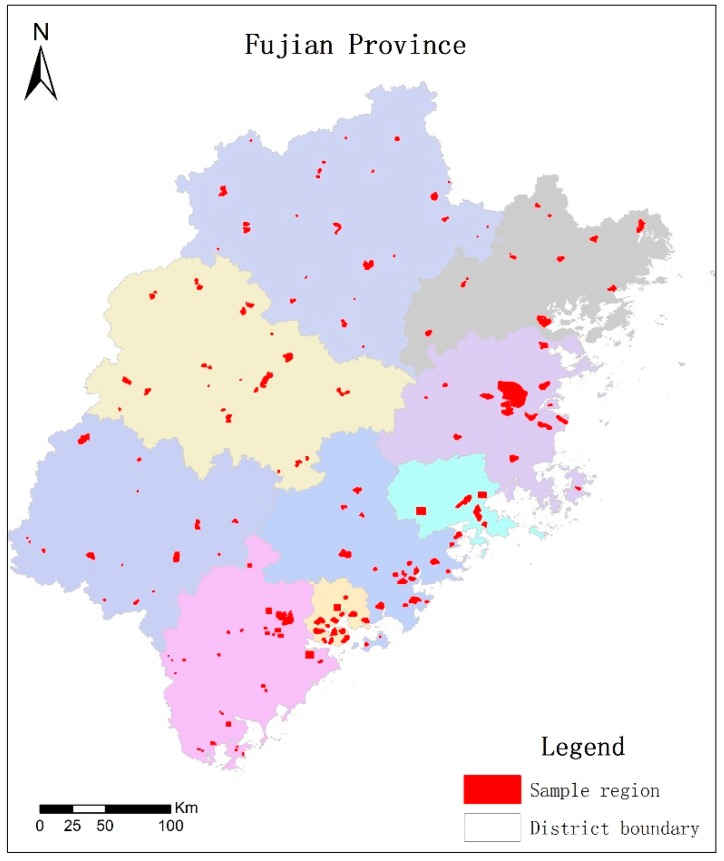
Typical regions of building samples in Fujian province of China. Images from Fujian province were collected and annotated manually to form the building outline dataset.

**Figure 7 sensors-19-00333-f007:**
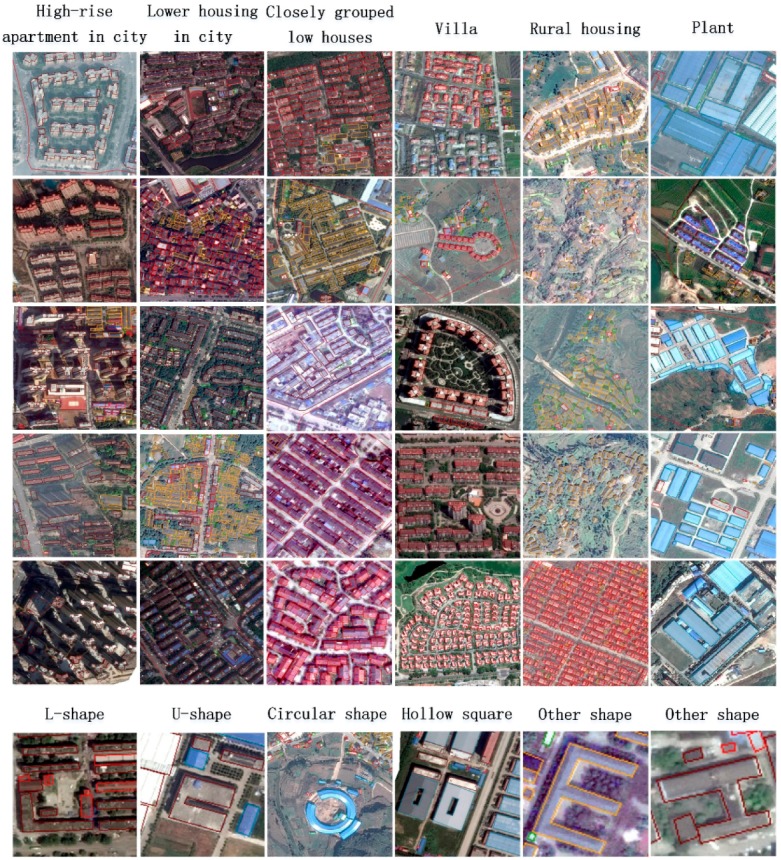
Building samples in our dataset. A variety of buildings with different shapes and architectural structures and functions are selected.

**Figure 8 sensors-19-00333-f008:**
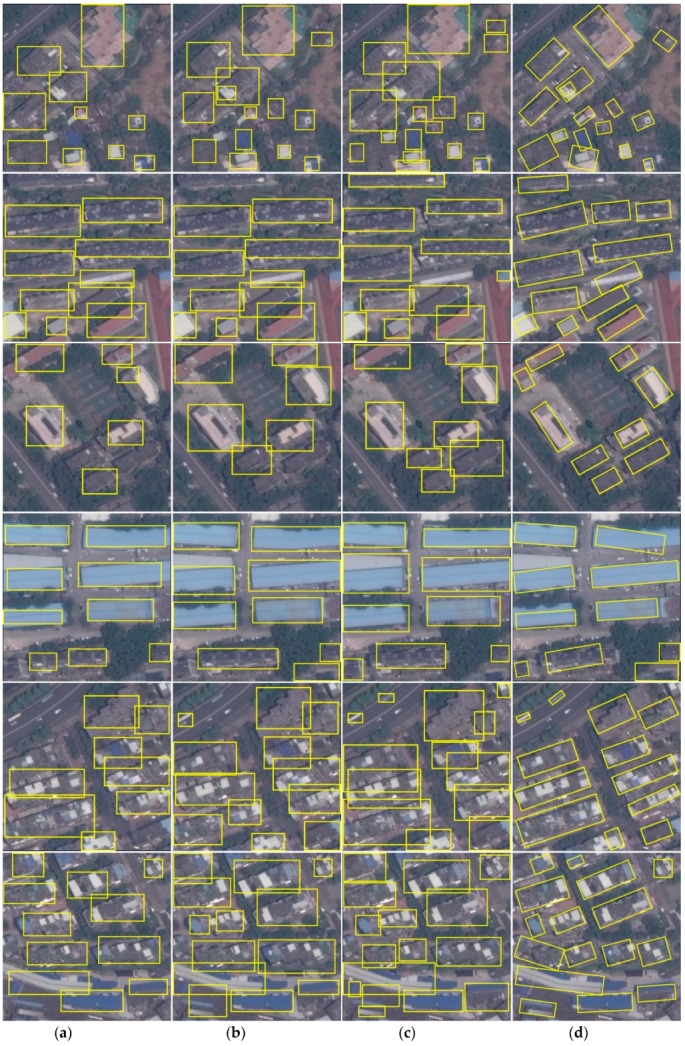
Comparison with other methods. (**a**) Faster R-CNN-VGG; (**b**) Faster R-CNN-ResNet101; (**c**) Mask R-CNN-ResNet101; (**d**) the proposed method.

**Figure 9 sensors-19-00333-f009:**
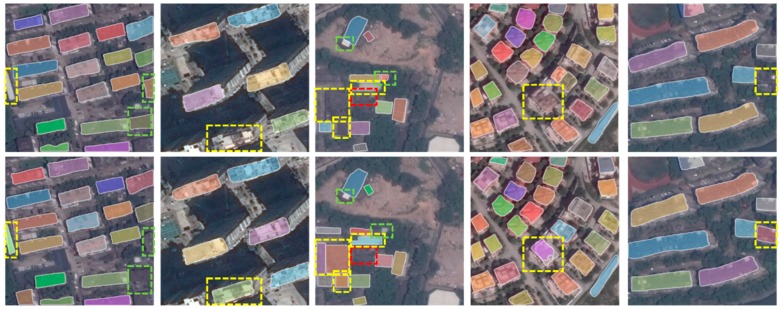
Building segmentation results. The first row shows results obtained by the Mask R-CNN-ResNet101, and the second row is the results of our method. Regions within yellow dashed lines are buildings successfully segmented by our method while missed by the Mask R-CNN-ResNet101. The green dashed lines indicate results obtained by the Mask R-CNN-ResNet101 while missed by our method. Red dashed lines are the results missed by both methods. As can been seen, our method obtains results better than Mask R-CNN-RnsNet101 in most case.

**Figure 10 sensors-19-00333-f010:**
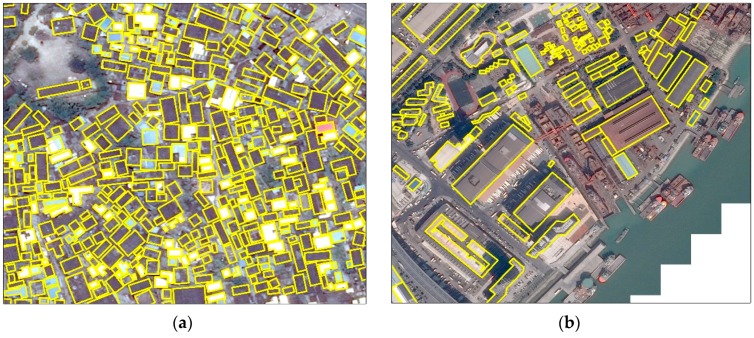
Various Building samples under complex backgrounds. (**a**) Building samples under complex background. There are streets, trees, bare land around these buildings, and the buildings are closely located to each other. (**b**) Building samples which are easily confused with ships in the same image. The ships have similar size, texture and shape to buildings.

**Table 1 sensors-19-00333-t001:** Comparisons with Faster R-CNN and Mask R-CNN on the Fujian dataset in terms of detection task.

Method	mAP
Faster R-CNN-VGG	0.6958
Faster R-CNN-ResNet101	0.8976
Mask R-CNN-ResNet101	0.8996
Proposed	0.9063

**Table 2 sensors-19-00333-t002:** Comparisons of the proposed method and Mask R-CNN-ResNet101 on the Fujian Dataset in terms of precession, recall, and *F*_1_-score.

Method	Precision	Recall	*F*_1_-score
Mask R-CNN-ResNet101	0.8627	0.8951	0.8786
Proposed	0.8720	0.9041	0.8878

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
