# Peer review of "Automatic Building Extraction from Google Earth Images under Complex Backgrounds Based on Deep Instance Segmentation Network"

_sensors, 2019, doi:10.3390/s19020333_

Round 1

Reviewer 1 Report

The authors present a well-written paper to detect rectangular building outlines from satellite-derived imagery based on deep instance segmentation network. I think the paper is very interesting, and only have few recommendations for improvement:

- It is not clear to the reader how the number of RBF modules impacts the performance and accuracy of the method.

- It is also unclear why the ROIs have to be aligned.

- Cross validation between learning and training sets would further improve the credibility of the method's accuracy.

Author Response

Dear sir/madam:

      Thank you very much for your careful reading and recommendations. We prepared the response of each point to explain the reasons. We hope that they are helpful to express our viewpoints. Thank you very much!

       Kind regards,

Qi Wen, Kaiyu Jiang, Wei Wang, Qingjie Liu, Qing Guo, Lingling Li, Ping Wang

9 January, 2019

Point 1: It is not clear to the reader how the number of RBF modules impacts the performance and accuracy of the method.

Response 1: Thanks for the comment. The benefits of RBF block are mainly in two-fold: firstly, the utilization of dilated convolutions enlarges receptive fields, allowing capturing information at a larger area with more context while keeping the same number of parameters; secondly, the inception-like architecture combines multiple convolutional kernels with varying sizes, which encodes richer multi-scale information, and thus leading to better and finer segmentation results. In this work, we intend to improve segmentation performance with modifications respect to the original Mask R-CNN as few as possible. Mask R-CNN has two successive convolutional layers in segmentation branch. We simply replace them with the RFB blocks. Normally, accumulating more RBF blocks would slightly improve the performance according to our experiment. For example, adding one more RBF block will improve about 0.09% accuracy, adding two RBF blocks will improve about 0.11% accuracy. However, when attaching more than three blocks, it will lead to unstable accuracy and make training more difficult. We observed a slightly drop of 0.03% when adding 3 blocks. We also add this explanation to section 2.2.2 in the manuscript.

Point 2: It is also unclear why the ROIs have to be aligned.

Response 2: Thanks for the comment. RoIAlign is an improvement on RoIPooling. In Faster R-CNN method, regional proposal network (RPN) generates RoI regions potentially containing objects.  These regions have different sizes. To speed up training and testing, Faster R-CNN applies RoIPooling to project the feature maps in RoI regions into fix-sized dimension. In forward pass, RPN generates a set of proposes, i.e. RoI regions, which may have floating number coordinates and then are rounded to integers (can be considered as a nearest neighbour interpolation) by RoIPooling. This operation will cause inaccurate localization and will become more serious when performing segmentation.  Suppose one input image has a size of 512×512 pixels, its feature map is 16×16×512 (32× downscaling), after RPN and RoIPooling, 0.1-pixel error in feature map will give rise to 3.2 pixel shifting in image domain, this phenomenon is called misalignment. To tackle this problem, RoIAlign is introduced. In each RoI region, the value of the four regularly sampled locations are computed directly through bilinear interpolation. Thus, avoid the misaligned problem.

Point 3: Cross validation between learning and training sets would further improve the credibility of the method's accuracy.

Response 3: Thanks for this suggestion. Cross validation has been widely used in early days to estimate how accurately a predictive model will perform in practice. In deep learning age, training a deep neural model will take days, e.g. it takes about 3 days to train the proposed method. Applying cross validation to assess one deep model will take weeks or even months, which is unwelcome in deep learning community due to its huge time costs. Since datasets used to train and test the deep learning models are large enough, it is widely believed that one validation on the test set could evaluate the performance and credibility of performance to an extent. In our paper, we split the dataset equally into two parts, considering there are 4 methods to be compared with, it will take about 30 days to conduct cross validation. We have confidence that even without cross validation, the experiments could prove the superiority of the proposed method.

Reviewer 2 Report

The authors have presented a pretty decent work, but it should be perfected before a possible publication in this journal. This reviewer has found the following issues that should be discussed or corrected by the authors:

Most of the acronyms in the text have not been defined.

Define R-CNN acronym also in the abstract.

L57-59 et seq.: Define acronyms in the text first time they appear.

L108: Add a new paragraph after "targets.", and same in L120 after "task."

L125: The first sentence is not necessary.

L133: Past tense incorrect here.

Figure 1: Font size must be legible. Maybe rearranging their elements should help.

Figures 2 and 3 are not mentioned in the text.

Figures 3, 4, 5: Delete initial "The". Is really necessary figure 5?

L168-169 et seq.: Don't use * as a "x" sign.

L254: Include units (pixels, meters?)

Figure 6: Improve picture quality. Do not use "we" here, and verb tenses are incorrect.

L272: Explain "with 12 GigaByte on board", it is not clear. Verb tense also incorrect.

L277: Change "Locating buildings" for "Building detection" directly, or delete the text in parenthesis.

L301: "inclined angle" does not sound proper.

Table 1: mAP has any units and/or range? Please include it. Same for parameters in Table 2.

Figures 8 and 9: Rectangles are barely seen, use another color to contrast them. You can use the same color and line width as in Fig. 10.

Discussion is too vague, it should be completed with references and actual aspects that made the proposed procedure better than the other analysed.

And again, please revise grammar.

Author Response

Dear sir/madam:

      Thank you very much for your careful reading and recommendations. We prepared the response of each point to present our revisions and explain the reasons. We hope that they are helpful to express our viewpoints. Thank you very much!

       Kind regards,

Qi Wen, Kaiyu Jiang, Wei Wang, Qingjie Liu, Qing Guo, Lingling Li, Ping Wang

9 January, 2019

Point 1: Most of the acronyms in the text have not been defined.

Point 2: Define R-CNN acronym also in the abstract.

Point 3: L57-59 et seq.: Define acronyms in the text first time they appear.

Response 1,2,3: we add the full name of Mask R-CNN acronym as “Mask Region Convolutional Neural Network” in the abstract. We add the full name of the following acronyms in the text:

R-CNN   Region Convolutional Neural Network

VGG      Visual Geometry Group Network

ResNet   Residual Network

FCN       Fully Convolutional Network

R-FCN   Region Fully Convolutional Network

YOLO    You Only Look Once

SSD        Single Shot Multibox Detector

SVM      Support Vector Machine

MRF       Markov Random Filed

In addition, AlexNet is mainly designed by Alex Krizhevsky, GoogleNet is designed by researchers in Google Company. Visual Geometry Group is affiliated to Department of Engineering Science, University of Oxford. All of the acronyms in the text are widely recognized by deep learning community and machine learning community.

Point 4: L108: Add a new paragraph after "targets.", and same in L120 after "task."

Response 4: we add a new paragraph after “targets” in line 108 and “task” in line 120.

Point 5: L125: The first sentence is not necessary.

Response 5: we delete the first sentence.

Point 6: L133: Past tense incorrect here.

Response 6: we change past tense as present tense.

Point 7: Figure 1: Font size must be legible. Maybe rearranging their elements should help.

Response 7: we enlarge the font size of figure 1. Now these texts are legible in the manuscript.

Point 8: Figures 2 and 3 are not mentioned in the text.

Response 8: we add the explanation of figures 2 and 3. “Figure 2 illustrates the outline and the minimum area bounding rectangle of buildings.” and “Figure 3 presents angle parameter                                                  of rotation bounding box.”

Point 9: Figures 3, 4, 5: Delete initial "The". Is really necessary figure 5?

Response 9: we delete initial “The” of figures 3, 4, 5. Figure 5 illustrates the structure of the segmentation branch of deep neural network. A detailed figure is necessary for readers to understand the structure definitely, otherwise, it is confusing for researchers to recurrent the experiment.

Point 10: L168-169 et seq.: Don't use * as a "x" sign.

Response 10: we change * as  in line 168-169.

Point 11: L254: Include units (pixels, meters?).

Response 11: we add “pixels” after “1000010000”.

Point 12: Figure 6: Improve picture quality. Do not use "we" here, and verb tenses are incorrect.

Response 12: we produce a new picture of figure 6 with resolution of 600 dpi. We revise the sentence as “Images from Fujian province were collected and annotated manually to form the building outline dataset”.

Point 13: L272: Explain "with 12 GigaByte on board", it is not clear. Verb tense also incorrect.

Response 13: The statement “with 12 GigaByte on board” means the memory of GPU is 12 GB. We have revised this to “with 12 GigaByte memory on board”. The verb tense has also been corrected.

Point 14: L277: Change "Locating buildings" for "Building detection" directly, or delete the text in parenthesis.

Response 14: We change “Locating buildings” for “Building detection”.

Point 15: L301: "inclined angle" does not sound proper.

Response 15: we change “inclined angle” as “rotation angle”.

Point 16: Table 1: mAP has any units and/or range? Please include it. Same for parameters in Table 2.

Response 16: mAP is the abbreviation of “mean Average Precision”. It is widely used in experiments to evaluate performance of object detection or multi-label image classification. Suppose a binary classification problem, for each testing sample we have a confidence score, ranging from 0 to 1, indicating probability of this sample being positive or negative category. Applying different thresholds, we have varying precisions and recalls, which can be plotted in the form of precision-recall curve. AP is averaging of precision with different thresholds, and is also area under precision-recall curve. mAP is mean AP for multi classes. Its range is from 0 to 1. And it has no unit.

Point 17: Figures 8 and 9: Rectangles are barely seen, use another color to contrast them. You can use the same color and line width as in Fig. 10.

Response 17: We have revised the Figures 8 and 9 and highlighted the boxes with yellow colour to make it more readable.

Point 18: Discussion is too vague, it should be completed with references and actual aspects that made the proposed procedure better than the other analysed.

Response 18: We have rewritten the discussion part, please refer the revised manuscript for more details. 

Round 2

Reviewer 2 Report

The authors have addressed all the issues and concerns laid out by this reviewer. The only points that should be reconsidered by the authors is the necessity of Fig. 5, as it does not aport much information to the reader, and to include value ranges in Tables 1 and 2. Anyway, this reviewer leaves that to the Editor's final decision.

Congratulations to the authors.